# Examination of the spatial-temporal variations in terrestrial water reserves and green efficiency of water resources in China's three northeastern provinces

**Yanying Wang**[1]*, **Xianzhi Wang**[1], **Longxue Zhao**[2]

**1** School of Geography and Tourism, Qilu Normal University, Jinan, China, **2** School of Geographical Sciences, Northeast Normal University, Changchun, Jilin, China

* 20176917@qlnu.edu.cn

## Abstract

Using technological advancements and analyzing urban water consumption patterns, this article employs GRACE satellite data and statistical records to conduct a comprehensive assessment and evaluation of water resource utilization efficiency across 34 prefecture-level cities in China's three northeastern provinces—Liaoning, Jilin, and Heilongjiang—over the period spanning from 2003 to 2020. By utilizing the sophisticated Super-SBM model, the study delves into the spatial and temporal variations in terrestrial water reserves and green water usage efficiency. Additionally, the Tobit model is introduced to investigate the influencing factors of water resource utilization efficiency. The primary findings of the study are outlined below: The spatial distribution of terrestrial water resources in the three northeastern provinces reveals a clear north-south gradient, with abundant resources in the northern regions and scarcity in the southern parts. Seasonal fluctuations, albeit present, are relatively modest, with higher water storage levels typically observed in spring and summer, and lower levels in autumn and winter. Regarding the static water use efficiency among the 34 prefecture-level cities, Panjin stands out with the highest efficiency, whereas Qiqihar ranks lowest. Notably, 91.18% of the cities exhibit medium to high efficiency levels, reflecting commendable performance in water utilization throughout the region. Almost half of the cities have experienced an improvement in their water use efficiency compared to the previous year, signaling a gradual enhancement in water utilization capabilities. The average total factor productivity across the three northeastern provinces stands at 1.012, representing an annual growth rate of 1.2%. The efficiency of water resource utilization in these provinces is intricately linked to the technological progress index. To enhance water resource utilization efficiency, it is imperative to introduce advanced technologies, increase research investments, and foster technological advancements.

**Data availability statement:** All relevant data are within the paper and its Supporting information files.

**Funding:** The author(s) received no specific funding for this work.

# 1 Introduction

Surface water reserves are a pivotal hydrological parameter, reflecting the cumulative effects of various inflows and outflows within a specific geographical area [1]. Understanding these dynamics is crucial for managing water resources sustainably. Water use efficiency (WUE) in ecosystems plays a fundamental role in modeling carbon and water cycles, as well as their intricate interplay [2]. This concept gained further prominence with the introduction of green efficiency of water resources by Sun et al. in 2017 [3], which most scholars have now integrated into the framework for analyzing water resources efficiency. Green efficiency of water resources emphasizes the balance between water resource inputs and the economic, social, and ecological outputs they generate, aiming for a harmonious triple-win situation for the economy, society, and ecology [3].

Research on water use efficiency can broadly be categorized into theoretical and empirical approaches [4]. Water resource utilization efficiency involves measuring and evaluating the relationship between water input and output to assess the effectiveness and efficiency of water resource use. This encompasses technical efficiency, which is determined by production techniques and process levels, and economic efficiency, influenced by factors such as economic development level and management proficiency [5]. Various techniques have been employed for empirically quantifying water use efficiency, including stochastic frontier analysis (SFA), entropy weights, and data envelopment analysis (DEA), along with derived models like non-radial directional distance function (NDDF), slacks-based measurement (SBM)-Malmquist, and MinDS [6,7]. As research evolved, the significance of nondesired outputs became apparent, leading to an increased adoption of SBM models in assessing water use efficiency. Today, SBM models are the foremost measurement approach and have been widely adopted in numerous studies [8–11].

Alterations in the global gravity field offer powerful insights into changes in terrestrial water storage [12]. The Gravity Recovery and Climate Experiment (GRACE), a collaborative effort between the National Aeronautics and Space Administration (NASA) and the German Space Flight Center (DLR), provides timely gravity field data that reflect the dynamic interactions between the atmosphere, terrestrial water, oceans, and the solid Earth [13]. Tiwari et al demonstrated the utility of GRACE data by using 11 months of observations to calculate water storage changes in the Mississippi River Basin, Amazon River Basin, and the Bay of Bengal region, highlighting the potential of Earth's gravity field data for monitoring regional-scale variations in water storage [14]. Since then, GRACE has become a widely accepted tool in both global and regional research on terrestrial water storage [15].

In this study, we utilize GRACE data spanning from January 2003 to October 2020 to infer terrestrial water storage in the three northeastern provinces of China. These data are then incorporated into the Super-SBM model to investigate water-use efficiency in these provinces and assess the contribution of total terrestrial water resources to water resource utilization. This research is particularly significant given the abundant water resources in the three northeastern provinces [16,17], which are vital for supporting regional economic development. However, recent population growth, misuse of water resources, and water pollution have disrupted their distribution, leading to challenges such as severe groundwater overexploitation, conflicts between agricultural water use and available water resources, and a decline in water purification capacity.

Despite the abundance of water resources in the northeastern provinces, there is currently no consensus among domestic scholars regarding the establishment of a comprehensive metric system for water resource utilization efficiency. Most scholars typically select input indicators based on capital, technology, labor, and water supply, with economic output serving

as a preferred output indicator, and water resource pollution identified as a nondesired output [18–20]. Considering the unique challenges and realities of urban water resource utilization in the northeastern provinces, this study establishes a water resource utilization efficiency evaluation indicator system that includes both natural and humanistic aspects as input indicators, per capita GDP as the expected output, and industrial wastewater discharge as the nonexpected output. This system takes into account the impact of nondesired outputs, making the evaluation results more objective and comprehensive.

Additionally, this study integrates terrestrial water storage as an input indicator, reflecting the importance of sustainable management and risk response in urban water resource utilization. This makes the evaluation system more holistic and integrated, providing a more accurate assessment of water resource utilization efficiency in the northeastern provinces.

The National Development and Reform Commission, alongside other relevant departments, has issued comprehensive guidelines aimed at bolstering the conservation and efficient utilization of water resources. These guidelines advocate for the refinement of water-saving system policies, the promotion of holistic water resource management, scientific allocation, comprehensive conservation, and circular utilization practices. They emphasize the vigorous advancement of water conservation initiatives in pivotal sectors such as agriculture, industry, and urban areas. Furthermore, the guidelines stress the importance of enhancing the utilization of unconventional water sources, fostering the development of water-saving industries, and constructing a society that prioritizes water conservation. It is envisioned that by the year 2030, the water-saving system, market regulation mechanisms, and technical support capabilities will undergo continuous enhancement, leading to a substantial improvement in water use efficiency and overall benefits.

The significance of this research lies in its focus on the municipal level in the northeastern provinces, where limited studies on water resource utilization efficiency exist. By elucidating the current status of water resource utilization efficiency, analyzing strengths and weaknesses, and exploring avenues for enhancement, this study aims to contribute to the ecological and sustainable development of the northeastern provinces. Furthermore, by conducting a targeted assessment of factors impacting water resource utilization efficiency, this research aims to inform more scientific and effective water resource management policies, ultimately enhancing the overall benefits of water resource utilization in the region and beyond.

## 2 Data and methods

### 2.1 Overview of the study area

Liaoning, Jilin, and Heilongjiang constitute the three eastern provinces, encom-passing a total of 36 prefectural and municipal administrative entities (Fig 1). This study was limited to 34 prefectural and municipal entities, as data for Yanbian Korean Autonomous Prefecture in Jilin Province and Daxinganling District in Heilongjiang Province were unavailable. Spanning an approximate area of 79.18×104 km$^2$, the three northeastern provinces experience brief, warm summers and prolonged, chilly winters, fostering a vast water network owing to substantial snowfall and its subsequent slow melt. The region is characterized by major rivers such as Heilongjiang, Songhua, Ussuri, Mudan, Yalu and Liao, which are primarily fed by meltwater from snow and ice, pre-cipitation, and groundwater.

This region, ideally situated amid mountains and water bodies, boasts advantageous conditions for agricultural cultivation and plays an essential role as a major grain-producing area in China. Concurrently, the trios of the eastern provinces are en-dowed with abundant mineral resources and represent the birthplace of heavy industry in contemporary China. The trajectory of industrial development in these three eastern provinces started at an earlier stage, and

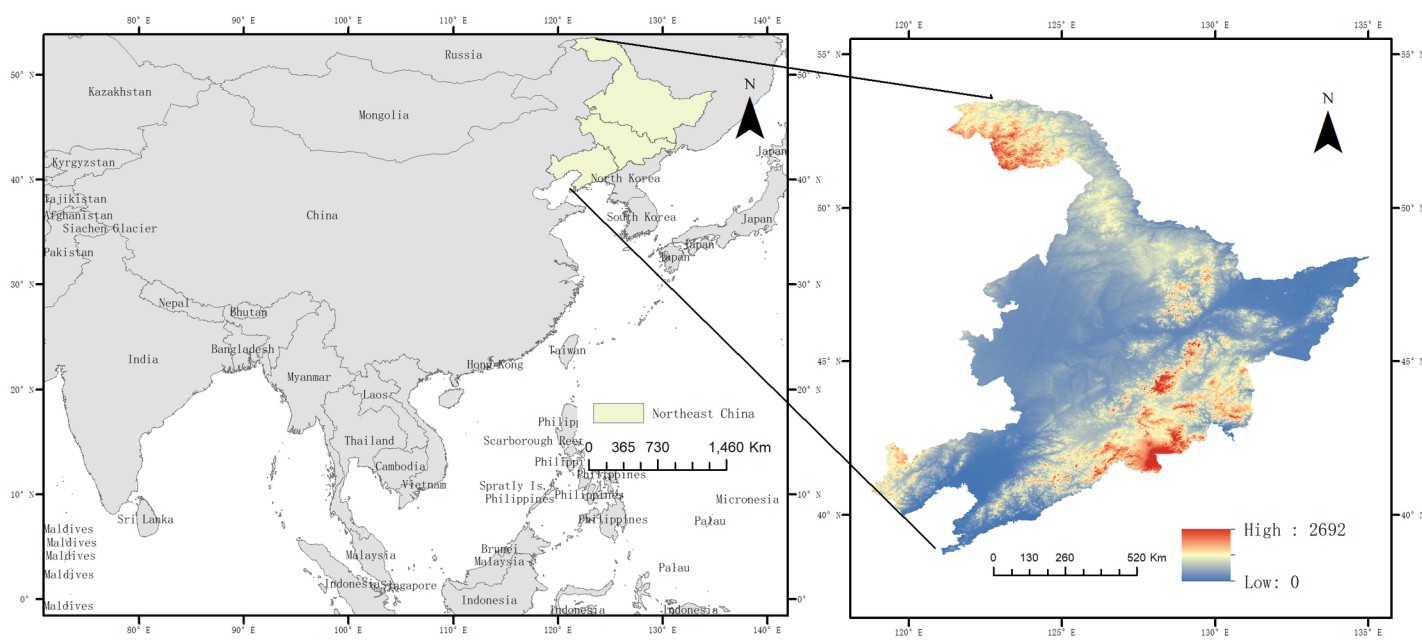

**Fig 1. Geographical overview of Northeast China.**

they are currently undergoing a phase of transition. Both the agriculture and industry sectors impose rigorous demands on the reserves and quality of water resources in these three eastern provinces.

## 2.2 Data processing

**2.2.1 GRACE data.** This study utilizes GRACE and GRACE-fo RL06 spherical harmonic coefficient monthly solution data furnished by the University of Texas between January 2003 and October 2020 and processes them appropriately. The specific processing steps are as follows:

(1) Execution of the spatial filtering and interpolation process

The GRACE gravity satellite's provided spherical harmonic coefficients are em-ployed for calculating the Earth's surface time-varying gravity field, delineated as follows:

$$\Delta\delta(\theta,\phi) = \frac{\alpha\rho_{\text{are}}}{3}\sum_{i=0}^{\infty}\sum_{m=0}^{l}\frac{2l+1}{1+k_l}\overline{P_{lm}}(\cos\theta)\times(\Delta C_{lm}\cos(m\phi)+\Delta S_{lm}\sin(m\phi)) \qquad (1)$$

where $\Delta\delta(\theta,\phi)$ is the change in mass density of the Earth's surface; $Q$ is the Earth's mean density; $\alpha$ is the Earth's mean radius; $\overline{P_{lm}}(\cos\theta)$ is the specified concatenated Legendre function; $\theta$ and $\phi$ are the Earth's core residual latitude and the Earth's core longitude, respectively; and $\Delta C_{lm}$ and $\Delta S_{lm}$_lm are the amount of change provided by GRACE.

Prior to processing the collected data from the GRACE satellite, initial preprocessing steps are essential [21]:

- Implement a geocentric correction for its 1-degree coefficients;
- Utilize Satellite Laser Ranging (SLR) to substitute the C20 coefficients within the gravity field model;

- Apply a blend of P3M6 decorrelation and sector filtering to the GRACE data to minimize the impact of high-frequency and correlation errors within a 300 km radius;
- Finally, the scale factor method was used to restore signals that were diminished during the filtering and degree correction processes.

Upon completion of the aforementioned four stages of data processing, we derived terrestrial water storage capacity (TWSC) gridded data specifically for the three north-eastern provinces. Nevertheless, data gaps exist due to inconsistencies between the GRACE satellites and the GRACE-FO mission. Consequently, to compensate for these gaps, we utilized a restructured Chinese surface water storage dataset rooted in 2003–2020 precipitation data. This dataset originated from the China Meteorological Data Network.

The formula applied for TWSC data reformation is provided below:

$$TWSC_{rec} = \beta P^T \tag{2}$$

where $TWSC_{rec}$ is the TWSC data after the complementary data, $P$ is the monthly precipitation data, $\beta$ is the calibration parameter for the long-term trend term, and $T$ is the seasonal calibration parameter.

(2) Derrending

Derrending analysis of GRACE data involves eliminating long-term trend signals from the original time series data. This procedure facilitates the erasure of incremental and decremental changes in the data's horizontal direction, thereby enabling a more effective examination of the time series data volatility variations.

**2.2.2 Precipitation data.** The precipitation data required for this study are sourced from the China Meteor-ological Science Data Sharing Service Network and the Zenodo website. There are a total of 77 meteorological stations surrounding the study area. The Gaussian regression (GPR) method based on meteorological station data was used to generate monthly precipitation data from 2003 to 2020 for the three northeastern provinces of China. The data accuracy is 1 km, the coordinate system is WGS84, and the grid unit is millimeters (mm).

**2.2.3 Data on water use efficiency variables.** Referring to the research of multiple scholars and considering the availability of data, based on the "China Urban Statistical Yearbook" and "China Urban Construction Statistics", six indicators including economic development level, industrial structure, various social economic expenditures, technological progress, opening up to the outside world, and education level were selected to explore the influencing factors of water resource utilization rate in the three provinces of Northeast China, as shown in Table 1.

## 2.3 Research methodology

**2.3.1 Super-SBM modelling.** Drawing upon the principle of "relative efficiency assessment", the DEA model, constructed by Charnes and Cooper, exhibits efficacy in assessing numerous deci-sion-making units utilizing identical types of inputs and outputs; its applicability extends to analysing frontier production functions that involve complex multi-input and mul-tioutput scenarios [22]. A DEA analytical technique—the SBM model—was proposed [23], grounded in slack variable metrics and characteristically nonradial as well as nonangular. However, this model does not exhibit a strict monotonic reduction correlated with varia-tions in the degree of slack in both inputs and outputs [24]; moreover, it lacks effectiveness in assessing and ranking decision-making units. The non-expected output Super-SBM model introduces slack variables into the objective function, which can avoid biases caused by

**Table 1. Evaluation index system for water resource utilization efficiency in the three Northeastern provinces.**

| Indicator | Indicator name | Indicator description | Unit |
|---|---|---|---|
| Input indicators | Terrestrial water storage capacity (TWSC) | Assessing the sustainability of water resource utilization | cm |
| | Precipitation depth | Measure the water storage capacity | mm |
| | Urban water supply population | Measuring per capita water resource utilization | Ten thousand people |
| | Urban water supply volume | Assessing natural water purification and urban water supply conditions | Hundred million cubic meters |
| | Wastewater treatment capacity | Measuring wastewater recycling capacity | Ten thousand tons |
| Output indicators | GDP per capita | Expected output | Ten thousand yuan |
| | Industrial wastewater discharge volume | Unexpected outcome | Ten thousand tons |

the relaxation of inputs and outputs and radial selection, and can solve the ranking problem among relatively efficient units [25], thereby improving the accuracy of the evaluation results for the efficient use of water resources. The specific calculation formula is as follows.

$$\rho = \min \frac{\frac{1}{m} \sum_{i=1}^{m} \frac{\overline{x_i}}{\overline{x_{l0}}}}{\frac{1}{s_1+s_2} \left( \sum_{r=1}^{s_1} \frac{\overline{y_r}}{y_{r0}} / y_{r_0}^g + \sum_{l=1}^{s_2} \frac{\overline{b_l}}{b_{l0}} \right)}$$

$$\text{s.t.} \begin{cases} \overline{x} \geq \sum_{j=1,\neq k}^{n} \lambda_j x_j \\ \overline{y}^g \leq \sum_{j=1,\neq k}^{n} \lambda_j y_j^g \\ \overline{y}^b \geq \sum_{j=1,\neq k}^{n} \lambda_j y_j^b \\ \overline{x} \geq x_0, \overline{y}^g \leq y_0^g, \overline{y}^b \geq \lambda_j y_0^b, \overline{y}^g \geq 0, \overline{\lambda}_j \geq 0 \end{cases} \quad (3)$$

where $\rho$ is the target efficiency value; $x$ denotes the slack variable of input indicators; $\overline{y}^g$ denotes the desired output; $\overline{y}^b$ denotes the slack variable of undesired output; $(\overline{x}, \overline{y})$ is the reference point of the decision variable; $m$ and $s$ are the number of input and output indicators, respectively; $\lambda$ is the vector of weights; $x_j, y_i^g$ and $y_j^b$ denote the values of the inputs, desired outputs, and undesired outputs of city $j$; and $x_0, y_0^g, y_0^b$ denote the total values of inputs, desired outputs, and nondesired outputs in the city, respectively. The urban water use efficiency is denoted by $\rho$. Higher values of $\rho$ indicate greater urban water use efficiency.

**2.3.2 Malmquist exponential modelling.** Despite the optimization of the SBM model through the Super-SBM model, its analysis is restricted to a static perspective. The Malmquist (M index) exponential model, introduced by Malmquist in 1953, facilitates the dynamic analysis of productivity [26]. There are divergent opinions in academia on the decomposition approach of the Malmquist index model, this paper utilizes the analytical technique suggested by Fare and his team. In a comparison between period t and period t + 1, alterations in production efficiency can be broken down into comprehensive technical efficiency index (EFFCH) and technological progress index (TECH). EFFCH can be further subdivided into pure tech-nical efficiency index (PECH) and scale efficiency index (SEC) [27]. The EFFCH can in-dicate the level of organization and management of regional water resources, as well as the capacity for comprehensive resource allocation across the entire region. TECH rep-resents

the development and progress of technologies related to water resource utilization [28]. The adoption of this decomposition method is consistent with the hypotheses pro-posed in this article.

The following are the corresponding formulas.

$$
\begin{aligned}
MI^{t,t+1} &= \left[ \frac{D^t\left(x^{t+1}, y^{t+1}\right)}{D^t\left(x^t, y^t\right)} \times \frac{D^{t+1}\left(x^{t+1}, y^{t+1}\right)}{D^{t+1}\left(x^t, y^t\right)} \right]^{1/2} \\
&= \frac{D^{t+1}\left(x^{t+1}, y^{t+1}\right)}{D^t\left(x^t, y^t\right)} \left[ \frac{D^t\left(x^t, y^t\right)}{D^{t+1}\left(x^t, y^t\right)} \times \frac{D^t\left(x^{t+1}, y^{t+1}\right)}{D^{t+1}\left(x^{t+1}, y^{t+1}\right)} \right]^{1/2} \\
&= EC^{t,t+1} \times TC^{t,t+1} \\
&= PEC^{t,t+1} \times SEC^{t,t+1} \times TC^{t,t+1}
\end{aligned}
\tag{4}
$$

In Eq (8), the input indicator vectors for time periods $t$ and $t + 1$ are represented as $x^t$ and $x^{t+1}$, respectively. Similarly, $y^t$ and $y^{t+1}$, respectively denote the output indicator vectors for these same time periods. MI denotes the variation in total factor productivity. $D^t$ symbolizes the efficiency value of the DMU within period $t$. $MI^{t,t+1} > 1$ suggests that the productivity of the DMU's water resources in period $t + 1$ exceeds that of period $t$. $EC^{t,t+1} > 1$ implies that the *DMU* performance in period $t$ is superior to that in period $t + 1$. Finally, $TC^{t,t+1} > 1$ indicates a shift in DMU production efficiency from period $t$ to $t + 1$, reflecting an enhancement in management performance.

**2.3.3 Spatial autocorrelation.** This study investigated the spatial correlation of water use efficiency across three northeastern provinces between 2003 and 2020 by employing the Moran index as the analytical tool, which is defined as follows:

$$
I = \frac{n \sum_{i=1}^{n} \sum_{j=1}^{n} W_{ij} \left(X_i - \overline{X}\right) \left(X_j - \overline{X}\right)}{\sum_{i=1}^{n} \sum_{j=1}^{n} W_{ij} \sum_{i=1}^{n} \left(X_i - \overline{X}\right)^2}
\tag{5}
$$

Here, $n$ refers to the sample size. The variables $X_i$ and $X_j$ represent the mobile populations of the $i$th and $j$th cities, respectively. The average mobile population across all cities is denoted by $\overline{X}$. Additionally, $W_{ij}$ is an element of the spatial weight matrix intended to assist in the positioning of mobile population data for comparative geospatial analysis in the geographic area being studied.

The previous formula represents the global spatial autocorrelation analysis. For an ofervation at a given site, we calculate the Moran's I statistic as follows:

$$
I_i(d) = Z_i \sum W_{ij} Z_j
\tag{6}
$$

where $Z_i$ and $Z_j$ are normalized patterns of observations.

(i) Tobit regression model

The Tobit model is a regression model in which the dependent variable for regression analysis is constrained, as the values derived from the Super-SBM model are invariably positive. The Tobit model typically applies when various conditions constrain the type of the dependent variable [29]. The prerequisites for implementing the Tobit model includes: (1) the two segments of the explanatory variables being distinct and (2) the random variable aligning with the joint normal distribution during the model assumptions, exemplified by the following

formula:

$$y^* = \beta X_i + \mu i$$

$$y_i = \begin{cases} y_i^* & \text{if } y_i^* > 0 \\ 0 & \text{if } y_i^* < 0 \end{cases} \tag{7}$$

The latent variable is represented as $y^*$, while the finite dependent variable is given by $y_i$. $X_i$ is utilized as the explanatory variable, with the correlation coefficient denoted by $\beta$. The random error is denoted as $\mu$, and the ith decision unit is denoted as $i$, Based on Eq (8), the regression assumptions are as follows:

$$y_{i,t} = \beta_0 + \beta_1 GDP_{i,t} + \beta_2 CY_{i,t} + \beta_3 OPEN_{i,t} + \beta_4 E_{i,t} + \beta_5 G_{i,t} + \varepsilon_{i,t} \tag{8}$$

In this study, $GDP_{i,t}$ refers to the growth rate of gross domestic product, and $CY_{i,t}$ represents the industrial structure. $OPEN_{i,t}$ signifies the level of openness to foreign trade, depicted by the ratio of export volume to GDP. $E_{i,t}$ stands for education level, represented by the proportion of enrolled students at all levels in the total population at year end. $G_{i,t}$ describes the various social economic expenditures, demonstrated by the share of general public budget expenditures in GDP. $\varepsilon_{i,t}$ is a disturbance term.

## 3 Spatial and temporal analysis of terrestrial water reserves

Fig 2 illustrates the spatial distribution of terrestrial water resources across the three eastern provinces, indicating that the northern regions have greater water resources than do the southern areas. The maximum value of the TWSC, 8.32, is located near the Qiqihar-Heihe border in Heilongjiang Province.

Time series analysis was conducted on the land water storage data of the three provinces in Northeast China (Fig 3). The seasonal variation of land water storage in Northeast China is quite apparent, with higher levels in spring and summer and lower levels in autumn and winter. Further decomposition of the time series of land water storage changes in Northeast China yielded Fig 4 [30]. It was observed that the mean of the Seasonal Adjustment Factor for land water storage changes in Northeast China fluctuated near zero, indicating that the seasonal fluctuations of land water storage in this region are not significant. The mean of the disturbance term was also near zero, sug-gesting the credibility of this result. The Trend Cycle Component reflects the long-term and cyclical changes in land water storage in Northeast China. Analysis revealed that this component shows no significant overall upward or downward trend but exhibits poor repeatability, suggesting a certain level of instability in water resources in the Northeast region. It is generally believed that there are four types of patterns in the numerical changes of time series: Secular trend (T), Seasonal Variation (S), Cyclical Variation (C), and Irregular Variation (I). After decomposition using SPSS, four factors are generated: Ir-regular Variation (I), Seasonally Adjusted Series (whose value equals T+C+I), Seasonal Adjustment Factor (whose value is equivalent to S), and Trend Cycle Component (whose value equals T+C).

## 4 Water use efficiency analysis

### 4.1 Static water use efficiency analysis

We applied a nondirected Super-SBM-based measurement to analyse the water use efficiency of 34 prefecture-level cities across three eastern provinces between 2003 and 2020 by calculating annual figures for each region. As depicted in Table 2, out of 612 static efficiency

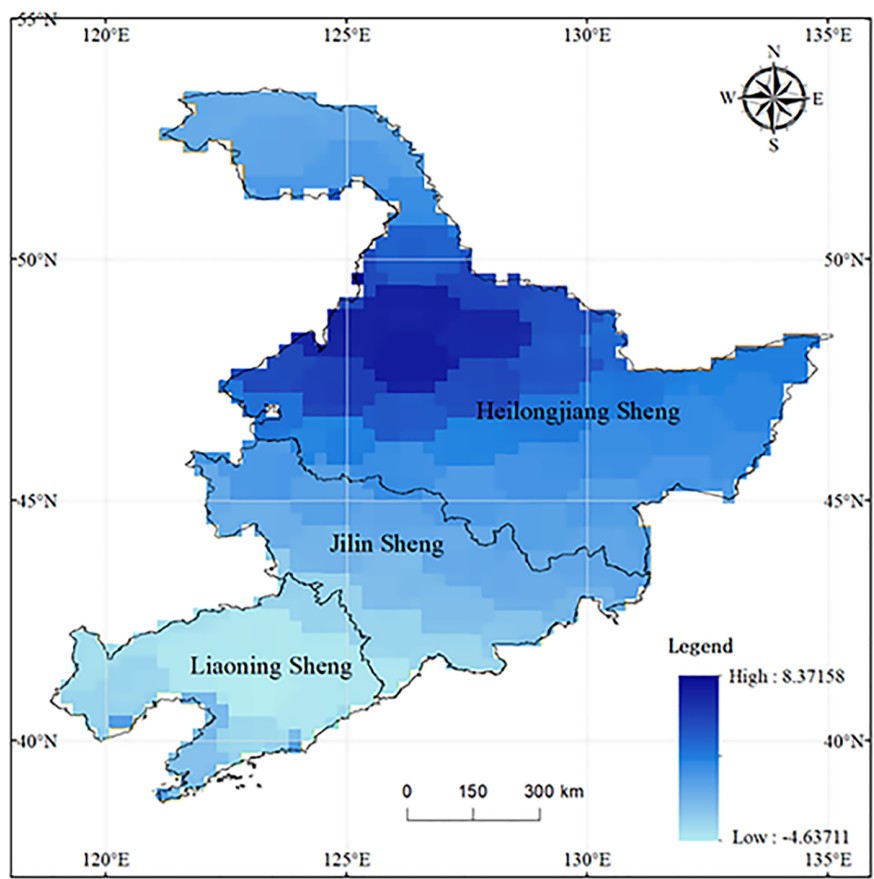

**Fig 2. Spatial distribution of the mean terrestrial water storage in the three northeastern provinces from 2003 to October 2020.**

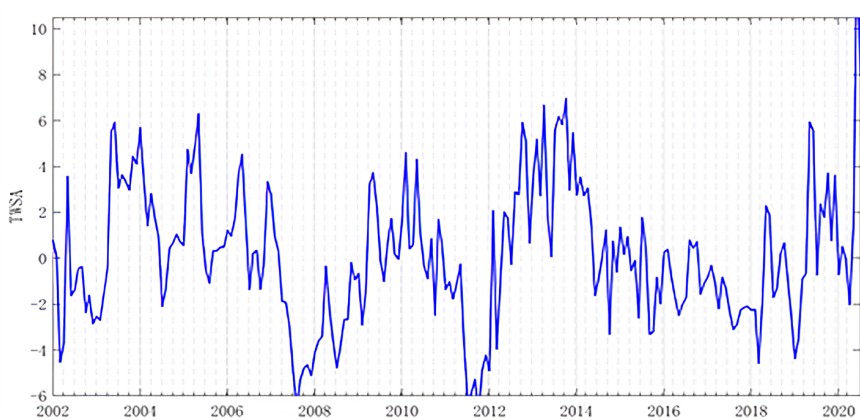

**Fig 3. Time series of terrestrial water resource reserves in the three northeastern provinces from January 2003 to October 2020.**

values, 248 values greater than or equal to 1 constituted 40.52% of the sample. These valid measurements provide valuable data for subsequent research.

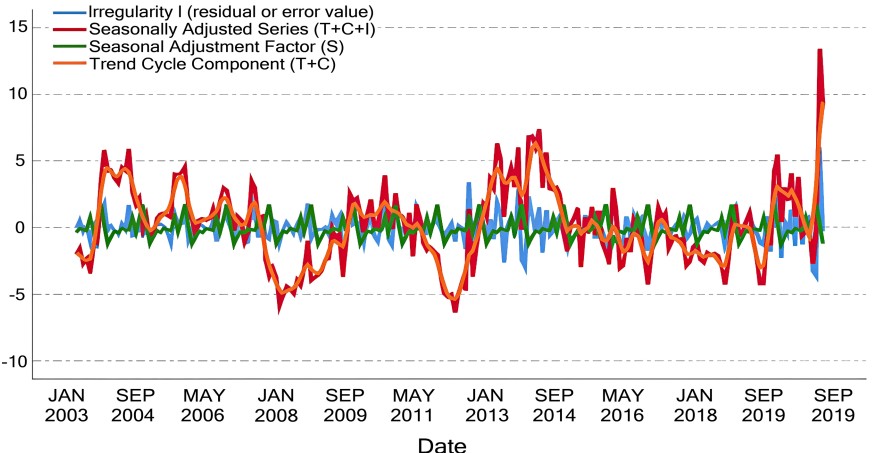

**Fig 4. Seasonal decomposition of the time series of terrestrial water resources in the three northeastern provinces from January 2003 to October 2020.**

**Table 2. Number of 34 prefecture-level cities in the three northeastern provinces with a static efficiency value of water resource utilization efficiency ≥ 1 from 2003 to 2020.**

| Year | 2003 | 2004 | 2005 | 2006 | 2007 | 2008 | 2009 | 2010 | 2011 |
|---|---|---|---|---|---|---|---|---|---|
| Number of valid values (in pieces) | 16 | 16 | 11 | 10 | 3 | 6 | 22 | 13 | 12 |
| **Year** | **2012** | **2013** | **2014** | **2015** | **2016** | **2017** | **2018** | **2019** | **2020** |
| Number of valid values (in pieces) | 13 | 12 | 14 | 18 | 17 | 14 | 18 | 17 | 16 |

**Table 3. The average value of the water resource utilization rate in 34 prefecture-level cities in the three northeastern provinces.**

| Region | Mean | Ranking | Region | Mean | Ranking |
|---|---|---|---|---|---|
| Panjin | 1.13 | 1 | Fuxin | 0.77 | 18 |
| Heihe | 1.11 | 2 | Hegang | 0.76 | 19 |
| Daqing | 1.09 | 3 | Suihua | 0.74 | 20 |
| Liaoyuan | 1.02 | 4 | Jiamusi | 0.72 | 21 |
| Baishan | 1.02 | 5 | Jixi | 0.71 | 22 |
| Yichun | 0.98 | 6 | FUshun | 0.70 | 23 |
| Baicheng | 0.95 | 7 | Shenyang | 0.69 | 24 |
| Songyuan | 0.92 | 8 | Dandong | 0.69 | 25 |
| Chaoyang | 0.90 | 9 | Anshan | 0.69 | 26 |
| Tonghua | 0.85 | 10 | Yingkou | 0.68 | 27 |
| Dalian | 0.83 | 11 | Jinzhou | 0.66 | 28 |
| Tieling | 0.83 | 12 | Huludao | 0.63 | 29 |
| Benxi | 0.82 | 13 | Mudanjiang | 0.63 | 30 |
| Shuangyashan | 0.82 | 14 | Changchun | 0.61 | 31 |
| Siping | 0.81 | 15 | Jilin | 0.54 | 32 |
| Qitaihe | 0.79 | 16 | Harbin | 0.52 | 33 |
| Liaoyang | 0.77 | 17 | Qiqihar | 0.44 | 34 |

Through a rigorous study, we obtained a robust understanding of static water use efficiency across the three eastern provinces and calculated the yearly mean static water efficiency for 34 prefecture-level cities [31,32] (refer to Table 3). Among the assessed cities, Panjin exhibited the maximum static water utilization efficiency, in stark contrast to Qi-qihar, which demonstrated the lowest efficiency. We established 1 and 0.6 as benchmarks for demarcating high,

medium, and low efficiency levels. Broadly, the three northeastern provinces demonstrate commendable water use efficiency, with 91.18% of the cities showcasing either medium or high efficiency levels. Nonetheless, certain areas, including Jilin, Harbin, and Qiqihar, display water utilization efficiencies lower than 0.6, indicating a pressing need for improvements in their water utilization efficiencies.

According to the spatial distribution (Fig 5), cities utilizing water resources effi-ciently are scattered geographically, potentially hindering the maximum exploitation of economies of scale. Regions at interprovincial borders, such as Daqing, Liaoyuan, and Baishan, are often home to cities with comparatively high water use efficiency. The northern region exhibits a diverse blend of areas with both high and low water use efficiency, displaying more pro-nounced regional disparities compared to its southern counterpart. Nonetheless, the southern region demonstrates a relative balance in overall water use efficiency, with a mean efficiency ranging from 0.6 to 1.

## 4.2 Dynamic water use efficiency analysis

The Super-SBM model fails to represent temporal variations in water use efficiency [33]. We employed DEAP 2.1 software to assess the dynamic water use efficiency in the three eastern provinces from 2003 to 2020, leveraging the chosen evaluation index system (refer to Table 4

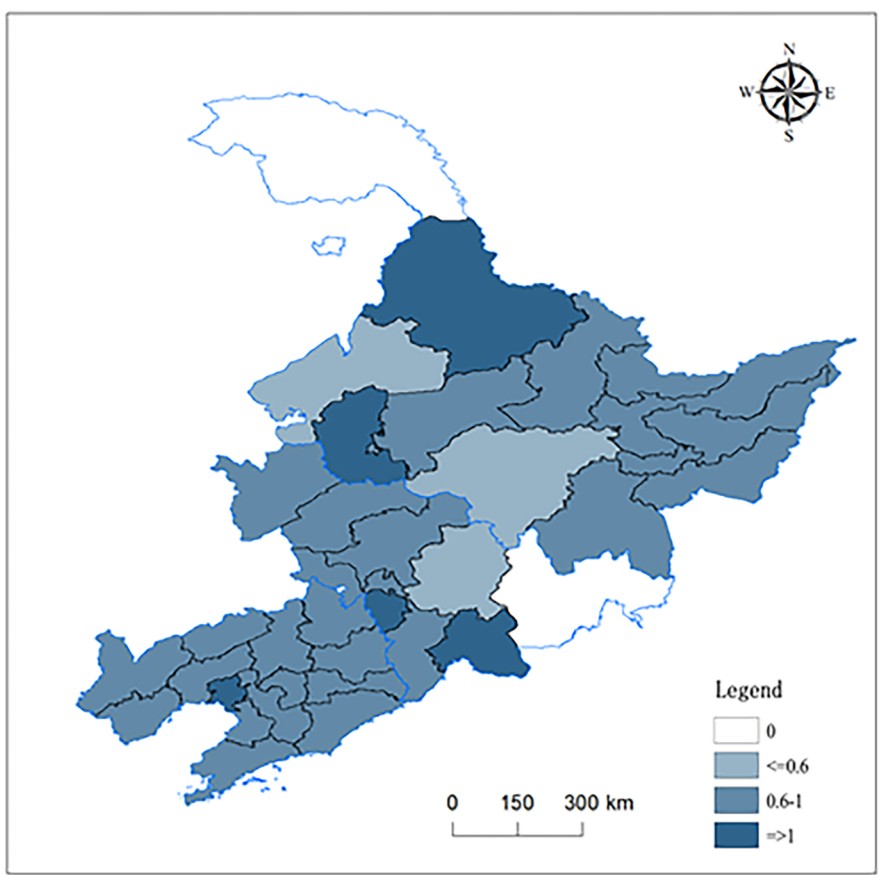

**Fig 5. Spatial distribution map of the mean value of water resource utilization in 34 prefecture-level cities in the three northeastern provinces from 2003 to 2020.**

**Table 4. Analysis of the dynamic water resource utilization rate in the three northeastern provinces from 2003 to 2020.**

| Year | 2003-2004 | 2004-2005 | 2005-2006 | 2006-2007 | 2007-2008 | 2008-2009 | 2009-2010 | 2010-2011 | 2011-2012 |
|---|---|---|---|---|---|---|---|---|---|
| Number of effective efficiency values (pcs) | 24 | 23 | 22 | 22 | 17 | 18 | 29 | 29 | 24 |
| Year | 2012-2013 | 2013-2014 | 2014-2015 | 2015-2016 | 2016-2017 | 2017-2018 | 2018-2019 | 2019-2020 | |
| Number of effective efficiency values (pcs) | 17 | 19 | 22 | 7 | 16 | 21 | 16 | 7 | |

for more details). Of the 578 calculated dynamic water efficiency scores, 333 exceeded 1, constituting 57.61% of the total sample size. This suggests that nearly half of the cities in the three eastern provinces have enhanced their water use efficiency compared to the preceding year, suggesting a steady improvement in the water utilization capacity of these prefecture-level cities.

We arrange and deconstruct the time series pertaining to the efficiency of dynamic water resources [34], as illustrated in Fig 6. The following content presents the results of this analysis:

- The overall dynamic water use efficiency of the prefecture-level cities in the three northeastern provinces is relatively good. 79.41% of the cities have a efficiency of water resource utilization greater than 1. Among the 34 prefecture-level cities, Shenyang has the highest dynamic water resource utilization efficiency, reaching 1.085, while Qiqihar has the lowest. 64.71% of the cities have dynamic water resource utilization rates surpassing the prefecture-level city average of 1.012. In the early years, there was greater fluctuation and disparity between EFFCH and TECH in the Northeastern provinces. However, in recent years, the gap between the two has narrowed. Consequently, the stability of dynamic water resource utilization efficiency has improved.
- From Table 5, it can be observed that the efficiency index of technological progress contributes more to the dynamic efficiency of water resource utilization. As seen from the

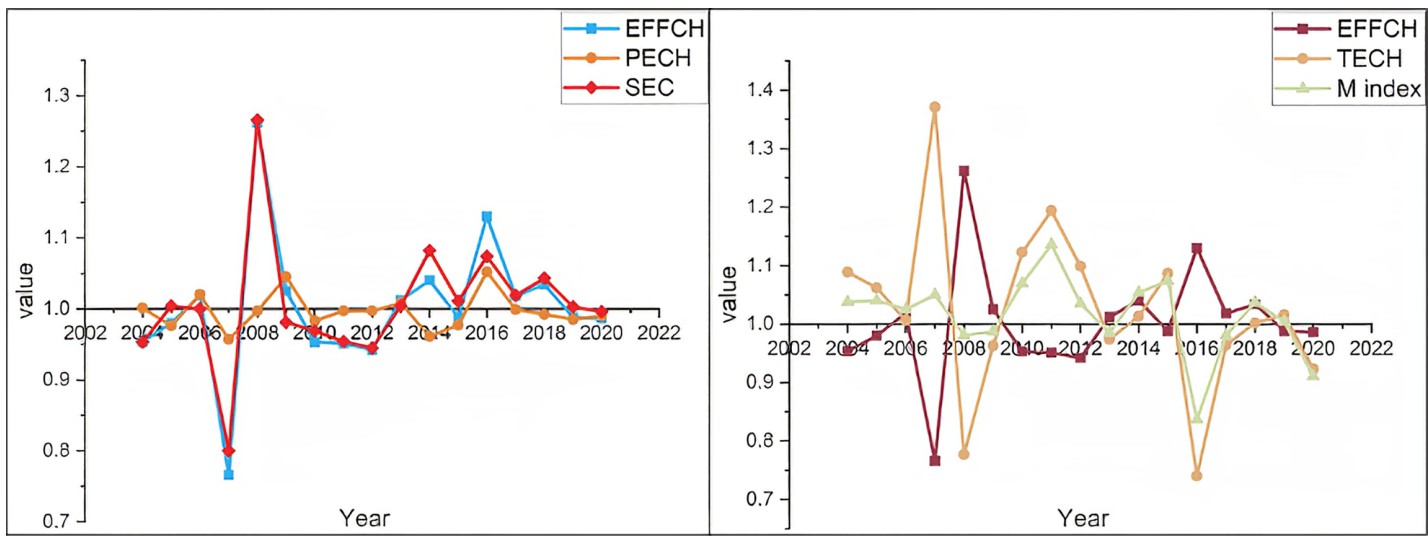

**Fig 6. Malmquist index and its decomposition for the three northeastern provinces.**

**Table 5. List of Malmquist Index decomposition items for prefecture-level cities in the three northeastern provinces.**

| Year | EFFCH | TECH | PECH | SEC | Malmquist index |
|---|---|---|---|---|---|
| 2003-2004 | 0.953 | 1.089 | 1.001 | 0.953 | 1.038 |
| 2004-2005 | 0.98 | 1.062 | 0.976 | 1.004 | 1.04 |
| 2005-2006 | 1.019 | 1.006 | 1.02 | 1 | 1.025 |
| 2006-2007 | 0.766 | 1.371 | 0.957 | 0.8 | 1.05 |
| 2007-2008 | 1.262 | 0.777 | 0.997 | 1.266 | 0.981 |
| 2008-2009 | 1.025 | 0.963 | 1.045 | 0.981 | 0.987 |
| 2009-2010 | 0.953 | 1.123 | 0.983 | 0.969 | 1.07 |
| 2010-2011 | 0.951 | 1.194 | 0.997 | 0.954 | 1.136 |
| 2011-2012 | 0.942 | 1.099 | 0.997 | 0.945 | 1.035 |
| 2012-2013 | 1.012 | 0.974 | 1.008 | 1.004 | 0.985 |
| 2013-2014 | 1.04 | 1.013 | 0.961 | 1.082 | 1.054 |
| 2014-2015 | 0.988 | 1.087 | 0.977 | 1.011 | 1.074 |
| 2015-2016 | 1.13 | 0.74 | 1.052 | 1.074 | 0.837 |
| 2016-2017 | 1.018 | 0.964 | 0.999 | 1.019 | 0.981 |
| 2017-2018 | 1.034 | 1.002 | 0.992 | 1.043 | 1.037 |
| 2018-2019 | 0.988 | 1.016 | 0.985 | 1.003 | 1.004 |
| 2019-2020 | 0.986 | 0.923 | 0.99 | 0.996 | 0.911 |
| mean | 0.998 | 1.014 | 0.996 | 1.002 | 1.012 |

decomposition index graph (Fig 6), the consistency between EFFCH and SEC is stronger. Therefore, efforts to improve the dynamic efficiency of water resource utilization in the Northeastern provinces should focus on enhancing technological efficiency. Moreover, if the goal is to enhance technological efficiency, greater emphasis should be placed on technology dissemination, rather than solely focusing on technological ad-vancement.

Considering that the subsequent momentum for urban sustainable development is often closely related to technological advancement, and based on the assumptions pre-sented in this paper, the study divides the 34 prefecture-level cities in Northeast China into four zones based on the average values of PECH and SEC (Fig 7). Additionally, tailored recommendations are proposed according to the characteristics of each region. The two prefecture-level cities of Shuangyashan and Yingkou are not located within any in-terval; they lie on the axes. The cities of Changchun and Harbin have relatively low PECH values compared to other cities, thus, in Fig 7, we specifically highlight these four special points.

In Zone I, both technical efficiency and scale efficiency are greater than the average techni-cal efficiency of prefecture-level cities as a whole. Water resource utilization is advantageous in the three provinces of the east, including 14 cities such as Shenyang, Liaoyang, and Heihe. These prefecture-level cities have good conditions and relatively high levels of water resource utilization. It is necessary for them to maintain their ad-vantages, keep pace with the pace of technological iteration and updates, and ensure that their advantages continue to be exerted.

In Zone II, the scale efficiency is greater than the average level of 34 prefecture-level cities, but the technical efficiency is lower than the provincial average. It includes four cities: Anshan, Changchun, Fushun, and Yichun. For these cities, the key to improving water resource utilization lies in enhancing their own technological strength and im-proving their operational and management systems.

In Zone III, both technical efficiency and scale efficiency are relatively low. Compared with other cities, they are at a relatively backwards level. The five cities included Daqing, Jiamusi, Mudanjiang, Harbin, and Qiqihar. It is difficult for them to catch up with the preceding cities, so they need to reflect on whether their water resource planning and municipal expenditures

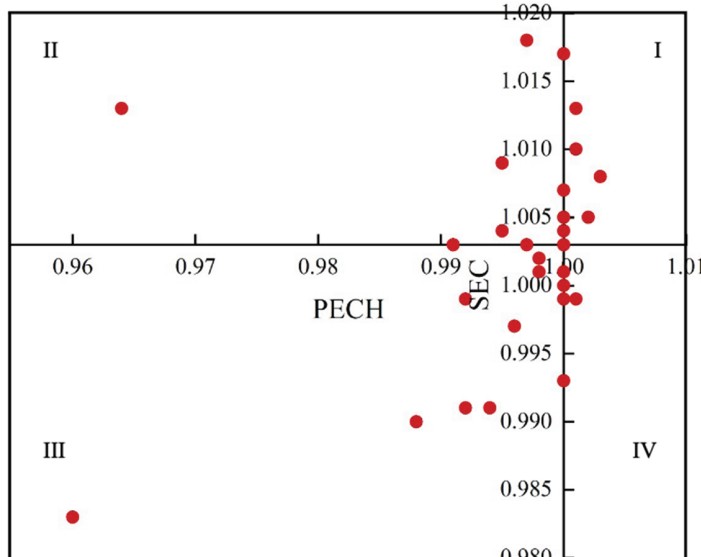

**Fig 7. The 34 prefecture-level cities in the three northeastern provinces are divided into four categories based on PECH and SEC.**

are reasonable and formulate a reasonable development model. At the same time, they should also pay attention to improving their own technology, learn from other cities or advanced cities in terms of water resource utilization in China, and adopt new technologies to promote the improvement of water resource utilization.

In Zone IV, the technical efficiency is relatively high, but the scale efficiency is lower than the average level of the three northeastern provinces. This means that these cities may have problems with water supply and drainage in some areas. It is recommended to consider increasing investment in these areas, promoting regional rectification, ration-alizing pricing and financial support, and improving the scale of water resource utili-zation.

## 4.3 Spatial autocorrelation analysis of water resource utilization efficiency

The spatial clustering of static water use efficiency in the Northeast Region was ex-plored using Geoda software (Table 6). Overall, the static water use efficiency in the three provinces of the Northeast Region did not exhibit significant spatial clustering every year. This study examined the spatial significance of the water resource utilization rates in the three provinces from 2003 to 2020 (Table 6) and revealed that the significance of recent spatial clustering decreased compared to that in previous years.

**Table 6. List of significant years in the global Moran index of the three northeastern provinces.**

| Years | 2004 | 2007 | 2011 |
|---|---|---|---|
| Moran's I | 0.223 | 0.196 | 0.153 |
| Z Score | 2.161 | 2.074 | 1.584 |
| P Value | 0.027** | 0.037** | 0.069* |

* Distinguish significance, * * * P < 0.001, indicating very significant.
** P < 0.05 is more significant, * P < 0.1 is significant.

At a confidence level of 95%, the local spatial autocorrelation of the three prov-inces in the Northeast Region was observed (Fig 8), and the changes in spatial clustering were examined at 5-year intervals. It was found that there was an im-provement in the spatial cluster-ing of water resource utilization, transitioning from low-low clustering to high-low clustering and high-high clustering areas. This indi-cates that the scale effect of cities is gradually taking effect. Some cities have recog-nized the importance of water resource utilization, continu-ously improving water re-source utilization rates, and generating regional effects that drive the improvement of water resource utilization efficiency in surrounding areas.

## 5 Analysis of influencing factors

A Tobit regression model was employed to investigate the factors influencing static water use efficiency in 34 prefecture-level cities located in the three northeastern provinces. Model 1 represents the regression model without considering control variables, while Model 2 includes time control variables in addition to Model 1 [29,35]. The outcomes are presented in Table 7, which indicates that the industrial structure, education level, overall economic

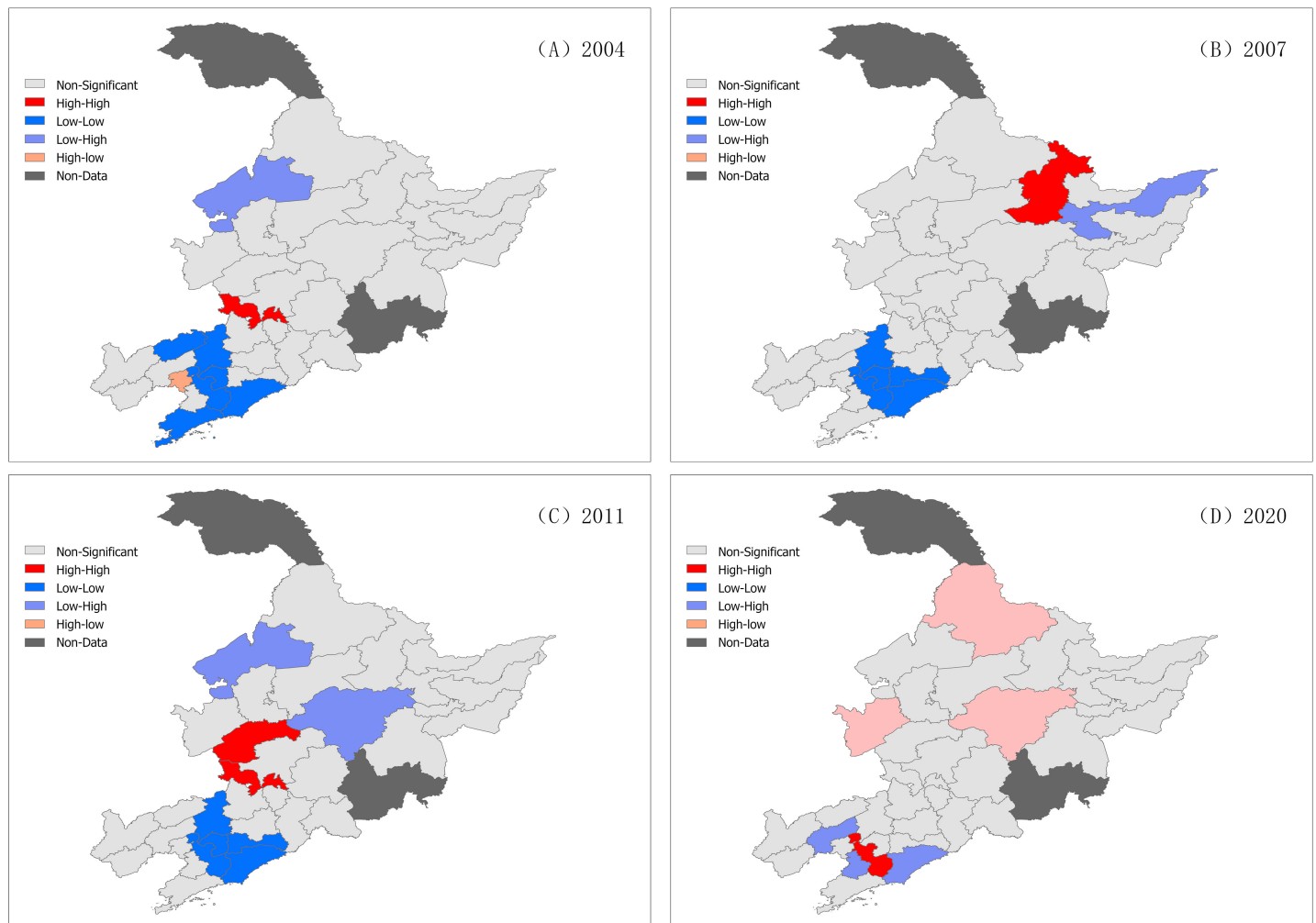

**Fig 8. Local spatial autocorrelation analysis of the three northeastern provinces in 2004 (A), 2007 (B), 2011 (C), and 2020 (D).**

**Table 7. Tobit regression analysis of the water resource utilization rate in the three northeastern provinces.**

| Model 1 | | | | | Model 2 | | | | |
|---|---|---|---|---|---|---|---|---|---|
| Et_Vt | Coef. | Std. Err. | z | P>z | Et_Vt | Coef. | Std. Err. | Z | P>z |
| GDP growth rate | −0.31*** | 0.09 | −3.31 | 0.001 | GDP growth rate | −0.07 | 0.11 | −0.68 | 0.49 |
| Industrial structure | 0.02 | 0.02 | 1.04 | 0.297 | Industrial structure | 0.04** | 0.02 | 2.84 | 0.01 |
| Open to the outside world | 0.00 | 0.00 | 0.29 | 0.768 | Open to the outside world | 0.00 | 0.00 | −0.80 | 0.43 |
| Educational level | 0.12 | 0.82 | 0.14 | 0.886 | Educational level | 3.86*** | 1.30 | 2.98 | 0.00 |
| Socioeconomic expenditures | −0.18 | 0.18 | −0.98 | 0.328 | Socioeconomic expenditures | −1.07*** | 0.20 | −5.23 | 0.00 |
| _cons | 0.81 | 0.09 | 9.02 | 0 | _cons | 0.52 | 0.15 | 3.50 | 0.00 |

* Distinguishes significance, ***P < 0.001 for highly significant.
**P < 0.05 for more significant, *P < 0.1 for more significant.

expenditures of society, and GDP growth significantly impact changes in static water use efficiency.

The studies by Qian and He [34], Mai et al. [36], and Ren et al. [37] support the positive impact of industrial structure on water use efficiency. The water utilization efficiency in the three northeastern provinces is closely linked to the technological progress index. The elimination of outdated waste industries and the active development of high-tech industries can improve water utilization efficiency.

The findings of Yang and Wu [38] as well as Zhao et al. [39] provide supporting evidence that education has a significant positive effect on water utilization. Education plays a crucial role in instilling water conservation among young students and enables higher-level students to apply their knowledge towards enhancing and optimizing water resource utilization techniques. As a result, this contributes to the overall enhancement of water resource utilization efficiency [29].

Socioeconomic expenditures have a significant deleterious effect on water use efficiency. To address the challenge of urban water utilization, it is necessary to augment financial expenditures and embrace the path of sustainable development. It would be beneficial to establish dedicated funds for enhancing water resources, as this would positively impact the lives of residents and foster the advancement of local industries in the three eastern provinces.

After accounting for control variables, it is observed that the GDP growth rate has a slight negative effect on water utilization. This implies that when the GDP growth rate is excessively rapid, the efficiency of water utilization in urban areas may decline. This can be attributed to the fact that an increase in GDP growth may result in a greater discharge of industrial wastewater, subsequently causing a reduction in water utilization. Therefore, while enhancing the input indicators of urban water utilization, emphasis should be placed on promoting the development of green industries and reducing the generation of unintended outputs.

## 6 Conclusion and suggestion

### 6.1 Conclusion

In this study, we evaluated the water resource efficiency of 34 prefectural-level cities in the three eastern provinces. We also analysed the spatial correlation of water resource use efficiency in the three northeastern provinces over different time periods. Additionally, we investigated the factors that contribute to variations in water resource use efficiency. The following section presents our findings and provides corresponding recommendations:

- The spatial distribution of terrestrial water resources in the three northeastern provinces of China, i.e., Liaoning, Jilin, and Heilongjiang, is characterized by a greater quantity in

the north and a lower quantity in the south, with more storage in spring and summer and less in autumn and winter. The seasonal differences are not significant. Overall, the utilization rate of static water resources in the three provinces is relatively good, with a high proportion of cities showing high and medium water resource utilization efficiency. However, there is still room for improvement in some areas. The distribution of cities with efficient water resource utilization is scattered, which hinders the realization of economies of scale. Southern cities have a more balanced distribution than do northern cities. The overall dynamic water resource efficiency of prefecture-level cities in the three northeastern provinces is good, and it is more consistent with the efficiency index of technological progress. Therefore, emphasis should be placed on the application of technology to improve water resource utilization.

- Different regions in the three northeastern provinces of China have varying factors that limit the development of water resources; therefore, it is necessary to improve their own water resource utilization according to local conditions. Of the total number of cities, 41.18% of the cities have advantageous water resource utilization efficiency, and 14.71% of the cities have achieved or surpassed the average level of water resource efficiency in the three provinces through scale efficiency. The key to improving water resource utilization lies in pure technological efficiency. A total of 29.41% of the cities need to improve their scale efficiency to enhance their own water resource utilization efficiency. This requires increased investment in funds and production supply scale. Additionally, 14.71% of the cities have lower water resource utilization efficiency than the provincial average. In these cases, increasing investment and expanding scale should be accompanied by a focus on improving their own technological capabilities.

- The spatial clustering of dynamic water resource utilization efficiency in prefecture-level cities in the three northeastern provinces is not strong, but over a longer time period, there has been a slight increase in spatial agglomeration. The main manifestation is the transition from low-low agglomeration to high-low agglomeration and high-high agglomeration. Areas with high water resource utilization efficiency have shown some growth, indicating a gradual increase in attention to water resource utilization.

- Water resource utilization efficiency is greatly influenced by industrial structure and education level. It is advisable to actively improve outdated industries and develop high-tech industries. At the same time, schools should implement water-saving awareness and behavior among students, cultivate water conservation consciousness, actively introduce professional talent, and contribute to the improvement of water resource utilization. Water resources are negatively affected by various social and economic expenditures. In the future, greater investment should be considered in improving water resource utilization efficiency, promoting the development of green industries, and pursuing sustainable development.

## 6.2 Limitations

- Due to battery issues, payload calibration errors, and long time intervals between GRACE and GRACE-FO missions, the time-varying gravity sequences observed by GRACE and GRACE-FO satellites are missing or interrupted, which affects the continuity and integrity of the results.

- The policy discussion on the impact on water resource efficiency is not in-depth enough. A more comprehensive policy analysis will provide valuable insights into the effectiveness of current measures and potential areas for improvement.

- This study mainly focuses on the three provinces in Northeast China. Expanding the geographical scope to include other regions or countries will improve the comparability and applicability of survey results.

### 6.3 Next step of research

- Multiple data fusion to improve research quality and investigate other factors that may affect water resource efficiency, such as climate change, technological progress, and cultural customs, in order to gain a more comprehensive understanding of this issue.
- By incorporating deep learning methods, future research on water resource utilization can be more scientific, providing a better and more comprehensive understanding of water resource efficiency and a scientific basis for the rational allocation of water resources.
- Expand the research scope to the national and even global level to compare and contrast water resource efficiency, gain a deeper understanding of policies that affect water resource efficiency, and identify effective strategies and potential areas for improvement.

## Author contributions

**Formal analysis:** Xianzhi Wang.

**Methodology:** Longxue Zhao.

**Writing – original draft:** Yanying Wang.

**Writing – review & editing:** Yanying Wang.

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
