## [Decision Letter · Decision Letter 0]

1 Nov 2024

PONE-D-24-22328Examination of the spatial-temporal variations in terrestrial water reserves and the efficiency of green water usage in China's three northeastern provincesPLOS ONE

Dear Dr. yanying,

Thank you for submitting your manuscript to PLOS ONE. After careful consideration, we feel that it has merit but does not fully meet PLOS ONE’s publication criteria as it currently stands. Therefore, we invite you to submit a revised version of the manuscript that addresses the points raised during the review process.

Dear Authors,

Thank you for submitting the revised manuscript " Examination of the spatial-temporal variations in terrestrial water reserves and the efficiency of green water usage in China's three northeastern provinces" to PLOS ONE.

Reviewers 3 and 4 have not recommended your paper and suggest major/minor revisions. The reviewer’s comments need to be addressed to improve the paper quality. I go through the revised manuscript and reviewers’ comments. I suggest you should address the reviewer’s comments and resubmit a revised version.

Yours sincerely

Dr. Malik Muhammad Akhtar

Academic Editor

PLOS ONE

We look forward to receiving your revised manuscript.

Kind regards,

Akhtar Malik Muhammad, PhD, Postdoc

Academic Editor

PLOS ONE

2. In your Methods section, please include additional information about your dataset and ensure that you have included a statement specifying whether the collection and analysis method complied with the terms and conditions for the source of the data.

4. We note that Figures 1 and 2 in your submission contain [map/satellite] images which may be copyrighted. All PLOS content is published under the Creative Commons Attribution License (CC BY 4.0), which means that the manuscript, images, and Supporting Information files will be freely available online, and any third party is permitted to access, download, copy, distribute, and use these materials in any way, even commercially, with proper attribution. For these reasons, we cannot publish previously copyrighted maps or satellite images created using proprietary data, such as Google software (Google Maps, Street View, and Earth). For more information, see our copyright guidelines: http://journals.plos.org/plosone/s/licenses-and-copyright.

1. You may seek permission from the original copyright holder of Figures 1 and 2 to publish the content specifically under the CC BY 4.0 license. 

Please upload the completed Content Permission Form or other proof of granted permissions as an ""Other"" file with your submission

Additional Editor Comments (if provided):

Reviewers' comments:

Reviewer's Responses to Questions

**Comments to the Author**

1. Is the manuscript technically sound, and do the data support the conclusions?

Reviewer #1: Yes

Reviewer #2: Partly

2. Has the statistical analysis been performed appropriately and rigorously? 

Reviewer #1: Yes

Reviewer #2: No

3. Have the authors made all data underlying the findings in their manuscript fully available?

Reviewer #1: Yes

Reviewer #2: Yes

4. Is the manuscript presented in an intelligible fashion and written in standard English?

Reviewer #1: Yes

Reviewer #2: Yes

5. Review Comments to the Author

Reviewer #1: This articel is designed to evaluate the spatial-temporal variations in terrestrial water reserves and the efficiency of green water usage in China's three northeastern provinces. The overall methodology and procedures were valid. However, the ABSTRACT should be re-summarized as there are no summarized conclusions and proper implications.

Reviewer #2: This paper uses GRACE data and statistical data to measure and evaluate the water resource utilization efficiency of 34 prefecture-level cities in the three northeastern provinces from 2003 to 2020 There are a few suggestions for further revision:

1.The entire paper should use a two end aligned format.

2.The title level is chaotic, for example, 1, 2, 3, etc. appear below 2.2.1..

3.The introduction section should elaborate on the importance and significance of this research in detail.

4.The first column of Table 2, Table 4 have two Year headers that need to be modified.

5.The clarity of most figures is low and needs to be improved.

6.The paper mentions the use of Tobit model to analyze the influencing factors of water resource utilization efficiency, but the application and analysis of formulas (7) and (8) in Tobit model are not used in the paper.

7.In the analysis of innovative points in the research content, there is no explanation of the innovative content, but rather a more accurate description of the research content is needed.

8.What is the basis for determining the evaluation index system?

9.I highly recommend to write limitations of the study and recommendation for future study at the end of conclusion.

10.The discussion on policies is not in-depth enough. It is recommended to include more policy analysis.

6. PLOS authors have the option to publish the peer review history of their article (what does this mean?). If published, this will include your full peer review and any attached files.

Reviewer #1: No

Reviewer #2: No

---

## [Author Response · Author response to Decision Letter 1]

18 Jan 2025

Dear Reviewer and Editor:

We sincerely appreciate the constructive comments provided by the reviewers and editors regarding our manuscript. Your insights and guidance have served as a ladder for our continuous improvement, significantly enhancing the quality of our work. We deeply regret the issues identified in the manuscript and assure you that we will internalize the advice from experts in our future research and studies, striving for higher levels of scholarly achievement. Below is a detailed response addressing each comment. The reviewers' comments are referenced using italicized font and numbering, while our responses are presented in regular font. Modifications and additions are highlighted in red for emphasis.

Recommendation 1: Please ensure that your manuscript meets PLOS ONE's style requirements, including those for file naming.

Response 1:

Thank you very much for your suggestion. I have made the modifications according to PLOS ONE's style requirements

Recommendation 2: In your Methods section, please include additional information about your dataset and ensure that you have included a statement specifying whether the collection and analysis method complied with the terms and conditions for the source of the data.

Response 2:

Thank you very much for your suggestion. The data is publicly available. And the data collection and analysis method complied with the terms and conditions set by the University of Texas for the use of these data

Recommendation 3: Please amend either the title on the online submission form (via Edit Submission) or the title in the manuscript so that they are identical.

Response 3:

Thank you very much for your suggestion. The modifications have been made as required.

Recommendation 4: We note that Figures 1 and 2 in your submission contain [map/satellite] images which may be copyrighted. All PLOS content is published under the Creative Commons Attribution License (CC BY 4.0), which means that the manuscript, images, and Supporting Information files will be freely available online, and any third party is permitted to access, download, copy, distribute, and use these materials in any way, even commercially, with proper attribution. For these reasons, we cannot publish previously copyrighted maps or satellite images created using proprietary data, such as Google software (Google Maps, Street View, and Earth).

Response 4:

Thank you very much for your suggestion. I have replaced the image with copyright issues.

Recommendation 5: This articel is designed to evaluate the spatial-temporal variations in terrestrial water reserves and the efficiency of green water usage in China's three northeastern provinces. The overall methodology and procedures were valid. However, the ABSTRACT should be re-summarized as there are no summarized conclusions and proper implications.

Response 5:

Thank you very much for your suggestions and advice. It is your help that enables me to continually improve. It is your guidance that has enabled us to continually improve. I have rewritten the abstract and summarized the conclusion again.

Recommendation 6: The entire paper should use a two end aligned format.

Response 6:

Thank you very much for your suggestions and advice. It is your help that enables me to continually improve. It is your guidance that has enabled us to continually improve. The article has been revised to align both ends.

Recommendation 7: The title level is chaotic, for example, 1, 2, 3, etc. appear below 2.2.1.

Response 7:

Thank you very much for your suggestions and advice. It is your help that enables me to continually improve. It is your guidance that has propelled our continuous improvement. The title number of the article has been modified.

Recommendation 8: The introduction section should elaborate on the importance and significance of this research in detail.

Response 8:

Thank you very much for your suggestions and advice. It is your help that enables me to continually improve. The introduction section of the article has been rewritten to supplement and elaborate on the importance and significance of the research.

Recommendation 9: The first column of Table 2, Table 4 have two Year headers that need to be modified.

Response 9:

Thank you very much for your suggestions and advice. The corresponding issues have been modified.

Recommendation 10: The clarity of most figures is low and needs to be improved.

Response 10:

Thank you very much for your suggestions and advice. The corresponding issues have been modified.

Recommendation 11: The paper mentions the use of Tobit model to analyze the influencing factors of water resource utilization efficiency, but the application and analysis of formulas (7) and (8) in Tobit model are not used in the paper.

Response 11:

Thank you very much for your suggestions and advice. The article has added a fifth part, which supplements the application and analysis of Tobit model.

Recommendation 12: In the analysis of innovative points in the research content, there is no explanation of the innovative content, but rather a more accurate description of the research content is needed.

Response 12:

Thank you very much for your suggestions and advice. The article has supplemented the innovation in the abstract and introduction sections.

Recommendation 13: What is the basis for determining the evaluation index system?

Response 13:

Thank you very much for your suggestions and advice. In section 2.2.3 of the article, the selection criteria for indicators have been supplemented.

Recommendation 14: I highly recommend to write limitations of the study and recommendation for future study at the end of conclusion.

Response 14:

Thank you very much for your suggestions and advice. The conclusion section of the paper supplements the limitations of the research and the next steps of research.

Recommendation 15: The discussion on policies is not in-depth enough. It is recommended to include more policy analysis.

Response 15:

Thank you very much for your suggestions and advice. In the introduction section, an analysis and discussion of policies have been added

---

## [Decision Letter · Decision Letter 1]

12 Jun 2025

PONE-D-24-22328R1Examination of the Spatial-Temporal Variations in Terrestrial Water Reserves and Green efficiency of water resources in China's Three Northeastern ProvincesPLOS ONE

Dear Dr. yanying,

Thank you for submitting your manuscript to PLOS ONE. After careful consideration, we feel that it has merit but does not fully meet PLOS ONE’s publication criteria as it currently stands. Therefore, we invite you to submit a revised version of the manuscript that addresses the points raised during the review process.

Dear Authors,

Thank you for submitting the revised manuscript " Examination of the Spatial-Temporal Variations in Terrestrial Water Reserves and Green efficiency of water resources in China's Three Northeastern Provinces" to PLOS ONE.

A reviewer has not recommended your paper and suggests minor revisions. The reviewer’s comments need to be addressed to improve the paper quality. I go through the revised manuscript and reviewers’ comments. I suggest you should address the reviewer’s comments very seriously and resubmit a revised version.

Yours sincerely

Dr. Malik Muhammad Akhtar

Academic Editor

PLOS ONE

We look forward to receiving your revised manuscript.

Kind regards,

Akhtar Malik Muhammad, PhD, Postdoc

Academic Editor

PLOS ONE

Journal Requirements:

Additional Editor Comments:

Dear Authors,

Thank you for submitting the revised manuscript " Examination of the Spatial-Temporal Variations in Terrestrial Water Reserves and Green efficiency of water resources in China's Three Northeastern Provinces" to PLOS ONE.

The reviewer-2 has not recommended your paper and suggests minor revisions. The reviewer’s comments need to be addressed to improve the paper quality. I go through the revised manuscript and reviewers’ comments. I suggest you should address the reviewer’s comments very seriously and resubmit a revised version.

Yours sincerely

Dr. Malik Muhammad Akhtar

Academic Editor

PLOS ONE

Reviewers' comments:

Reviewer's Responses to Questions

**Comments to the Author**

1. If the authors have adequately addressed your comments raised in a previous round of review and you feel that this manuscript is now acceptable for publication, you may indicate that here to bypass the “Comments to the Author” section, enter your conflict of interest statement in the “Confidential to Editor” section, and submit your "Accept" recommendation.

Reviewer #1: All comments have been addressed

Reviewer #2: All comments have been addressed

2. Is the manuscript technically sound, and do the data support the conclusions?

Reviewer #1: Yes

Reviewer #2: Yes

3. Has the statistical analysis been performed appropriately and rigorously? 

Reviewer #1: Yes

Reviewer #2: Yes

4. Have the authors made all data underlying the findings in their manuscript fully available?

Reviewer #1: Yes

Reviewer #2: Yes

5. Is the manuscript presented in an intelligible fashion and written in standard English?

Reviewer #1: Yes

Reviewer #2: Yes

6. Review Comments to the Author

Reviewer #1: (No Response)

Reviewer #2: 1.I did not see any traces of modification in the file “Revised Manuscript with Track Changes”

2.There is no figure 8 in the uploaded picture.

7. PLOS authors have the option to publish the peer review history of their article (what does this mean?). If published, this will include your full peer review and any attached files.

Reviewer #1: No

Reviewer #2: No

---

## [Author Response · Author response to Decision Letter 2]

18 Jun 2025

Dear Reviewer and Editor:

We sincerely appreciate the constructive comments provided by the reviewers and editors regarding our manuscript. Your insights and guidance have served as a ladder for our continuous improvement, significantly enhancing the quality of our work. We deeply regret the issues identified in the manuscript and assure you that we will internalize the advice from experts in our future research and studies, striving for higher levels of scholarly achievement. Below is a detailed response addressing each comment.

Recommendation 1: I did not see any traces of modification in the file “Revised Manuscript with Track Changes”.

Response 1:

Thank you very much for your suggestion. There was an error in the annotation of my previous revised manuscript, so I have re uploaded the file 'Revised Manuscript with Track Changes'.

Recommendation 2: There is no figure 8 in the uploaded picture.

Response 2:

Thank you very much for your suggestion. I'm very sorry, due to my negligence, my image 8 was lost during uploading. I have uploaded it again.

---

## [Decision Letter · Decision Letter 2]

7 Jul 2025

Examination of the Spatial-Temporal Variations in Terrestrial Water Reserves and Green efficiency of water resources in China's Three Northeastern Provinces

PONE-D-24-22328R2

Dear Authors,

We’re pleased to inform you that your manuscript has been judged scientifically suitable for publication and will be formally accepted for publication once it meets all outstanding technical requirements.

Kind regards,

Akhtar Malik Muhammad, PhD, Postdoc

Academic Editor

PLOS ONE

Additional Editor Comments (optional):

Dear Authors,

Reviewers and Academic Editor are satisfied the modifications and revised version of the manuscript. The resolution of all figures should be improved to meet standard quality requirements.

The paper is accepted for the publication and further process.

Congratulations to authors.

Thanks to submit your research work to “PLOS ONE”

Dr. Malik Muhammad Akhtar

Academic Editor

PLOS ONE

Reviewers' comments:

Reviewer's Responses to Questions

**Comments to the Author**

1. If the authors have adequately addressed your comments raised in a previous round of review and you feel that this manuscript is now acceptable for publication, you may indicate that here to bypass the “Comments to the Author” section, enter your conflict of interest statement in the “Confidential to Editor” section, and submit your "Accept" recommendation.

Reviewer #2: All comments have been addressed

2. Is the manuscript technically sound, and do the data support the conclusions?

Reviewer #2: Yes

3. Has the statistical analysis been performed appropriately and rigorously? 

Reviewer #2: Yes

4. Have the authors made all data underlying the findings in their manuscript fully available?

Reviewer #2: Yes

5. Is the manuscript presented in an intelligible fashion and written in standard English?

Reviewer #2: Yes

6. Review Comments to the Author

Reviewer #2: I am satisfied with the author's modification, but there are still some problems with the pictures, such as Figure 4 and Figure 8, the clarity of pictures is low, resulting in some distortion of the text.

7. PLOS authors have the option to publish the peer review history of their article (what does this mean?). If published, this will include your full peer review and any attached files.

Reviewer #2: No
